# High SUVmax Is an Independent Predictor of Higher Diagnostic Accuracy of ROSE in EBUS-TBNA for Patients with NSCLC

**DOI:** 10.3390/jpm12030451

**Published:** 2022-03-13

**Authors:** Ying-Yi Chen, Hsin-Ya Huang, Chi-Yi Lin, Kuan-Liang Chen, Tsai-Wang Huang

**Affiliations:** 1Graduate Institute of Medical Science, National Defense Medical Center, Taipei 114, Taiwan; chi-wang@yahoo.com.tw; 2Division of Thoracic Surgery, Tri-Service General Hospital, National Defense Medical Center, Taipei 114, Taiwan; 3Division of General Surgery, Changhua Christian Hospital, Changhua 500, Taiwan; addgujfjui@mail.ndmctsgh.edu.tw; 4Division of Thoracic Surgery, Department of Surgery, Armed Forces Taoyuan General Hospital, Taoyuan 325, Taiwan; thanksthegod@gmail.com; 5Division of Thoracic Surgery, Department of Surgery, Zuoying Branch of Kaohsiung Armed Forces General Hospital, Kaohsiung 813, Taiwan; doc20771@mail.ndmctsgh.edu.tw

**Keywords:** non-small cell lung cancer, endobronchial ultrasound, rapid on-site evaluation

## Abstract

Introduction: This study aimed to verify the predictors of the diagnostic accuracy of rapid on-site evaluation (ROSE) in endobronchial ultrasound-guided transbronchial needle aspiration (EBUS-TBNA) among patients with non-small cell lung cancer (NSCLC). Methods: We retrospectively reviewed consecutive patients with NSCLC who underwent EBUS-TBNA for staging or diagnosis at our hospital from June 2016 to June 2018. The patients were divided into two groups—those with a correct diagnosis and an incorrect diagnosis after ROSE. Kaplan–Meier plots and log-rank tests were used to estimate outcomes. Results: A total of 84 patients underwent EBUS-TBNA for staging and diagnosis. Sixty patients with demonstrated malignant mediastinal lymph nodes were enrolled. In the univariate analysis, lymph nodes < 1.5 cm (HR = 3.667, *p* = 0.031) and a SUVmax > 5 (HR = 41, *p* = 0.001) were statistically significant for diagnostic accuracy of ROSE. In the multivariate Cox regression analysis, only a SUVmax > 5 (HR = 20.258, *p* = 0.016) was statistically significant. Conclusions: A SUVmax > 5 is an independent predictor of higher diagnostic accuracy of ROSE in EBUS-TBNA in patients with NSCLC with malignant mediastinal lymph nodes. Therefore, ROSE in patients with a SUVmax < 5 might not be reliable and requires further prudent assessment (more shots or repeated biopsies at mediastinal LNs) in clinical practice.

## 1. Introduction

Abnormal intrathoracic lymph nodes (LNs) are usually selected for biopsy on the basis of a short axis of more than 1 cm by contrast-enhanced computed tomography (CT) of the chest or increased 18F-fluorodeoxyglucose (FDG) uptake by position emission tomography (PET)-CT. Endobronchial ultrasound-guided transbronchial needle aspiration (EBUS-TBNA) has revolutionized intrathoracic disease diagnosis and lung cancer staging and is a minimally invasive and pivotal procedure in invasive thoracic workup [1,2]. Due to substantial improvements in molecular medicine for non-small cell lung cancer (NSCLC), EBUS-TBNA is now an efficient method that provides adequate samples to test for epidermal growth factor receptor (EGFR) mutations and anaplastic lymphoma kinase gene arrangement following routine histopathology and immunohistochemistry (IHC)-based subtyping [3]. Although EBUS sonographic factors, such as gray scale, blood flow Doppler, and elastography, can be used to predict malignant and benign intrathoracic LNs, tissue samples should still be obtained to confirm the diagnosis [1,4]. Rapid on-site evaluation (ROSE) of EBUS-TBNA is a well-established tool that improves the procedure yield if performed by a cytopathologist or an experienced cytotechnologist and reduces additional biopsies while preserving the diagnostic yield [5]. However, ROSE still has some limitations [6], and many reports have revealed less than 100% diagnostic accuracy [7,8,9]. To our knowledge, no studies have investigated the predictors of the diagnostic accuracy of ROSE in EBUS-TBNA. The diagnostic accuracy of ROSE plays a particularly important role in clinical practice, especially in nodal staging for immediately intraopeartive treatment decision making. Therefore, this study aimed to correlate the results of ROSE and to determine the predictors of the diagnostic accuracy of ROSE.

## 2. Methods

### 2.1. Patient Selection and Study Design

The database of the Thoracic Surgery Division of Tri-Service General Hospital, National Defense Medical Center, Taipei, Taiwan, was searched for patients who underwent EBUS-TBNA for nodal staging and diagnosis of NSCLC between June 2016 and June 2018. The medical records of the study population were reviewed and evaluated. Patients who underwent EBUS-TBNA were suspected to have intrathoracic lymph node metastases on the basis of enlargement (short axis > 10 mm), as visualized by CT or F-18 fluorodeoxyglucose (FDG) uptake ≥ the standard uptake value of 3.5 by PET. All LNs in the thorax and extrathoracic regions with a SUVmax > 3.5 were considered positive, unless they showed high attenuation (>70 HU) or benign calcification (central nodular, laminated, popcorn, or diffuse), as seen on the soft-tissue window of CT images [10,11,12]. The 60 patients with malignant LNs and the 24 patients with benign LNs were proved by the final pathologies or further follow-up in clinical conditions. This study was approved by the Institutional Review Board of Tri-Service General Hospital, National Defense Medical Center (2-108-05-089). Written informed consent for bronchoscopy was obtained from all patients; additional informed consent for this study was waived due to the retrospective design of the chart review through which clinical history and diagnostic results were obtained.

### 2.2. Mediastinal Lymph Node Sampling

Patients under general anesthesia underwent EBUS-TBNA in an operating room with 8.0 mm endotracheal intubation. After white-light bronchoscopy was performed with a tracheal tube, the target LNs and peripheral vessels were examined by EBUS using a linear array ultrasonic bronchoscope (BF-UC180F-OL8; Olympus Ltd., Tokyo, Japan). The diameters of the target LNs were measured and recorded in the frozen ultrasound images. A dedicated 22G needle was used for aspiration (NA-201SX-4022; Olympus Ltd., Tokyo, Japan). We recommended that at least three needle aspirations be performed for each target lesion, and the number of passes was approximately 20 to 30 times [13,14]. All procedures were performed by experienced bronchoscopists. Needle aspiration of the largest accessible LNs was guided by ultrasound. An internal stylet was removed after the initial puncture, after which negative pressure was applied with a syringe to obtain histological cores and cytological specimens. The aspirated material was smeared onto glass slides and was then fixed in 95% alcohol. Papanicolaou staining and light microscopy were also performed by an independent cytopathologist. Histological cores were formalin-fixed and stained with hematoxylin and eosin. Immunohistochemistry was also performed when necessary. Biopsies from stations 4R, 7, and 4L were routinely obtained.

### 2.3. Statistical Analyses

Univariate and multivariate analyses were used to assess the independent risk factors for the diagnostic accuracy of ROSE in EBUS-TBNA. A *t*-test was used to compare continuous variables, while the chi-squared test or Fisher’s exact test was used to compare categorical variables when appropriate. Significance was indicated at *p* < 0.05, and all analyses were two-sided. Significant variables in the univariate analysis or those deemed clinically important were then entered in a multivariable logistic regression model. The IBM SPSS Statistics for Windows software package (ver. 20.0; IBM Corp., Armonk, NY, USA) was used for the data analysis.

## 3. Results

### 3.1. Demographic Characteristics of the Enrolled Patients

In all, 723 patients with NSCLC were eligible for this retrospective cohort study, and of them, 84 underwent EBUS-TBNA for staging and diagnosis from June 2016 to June 2018. Of those patients, we investigated 60 patients with demonstrated malignant mediastinal LNs to determine the diagnostic accuracy of ROSE. The confusion matrix showed the positive predictive rate was 97.67%, the sensitivity was 70%, the specificity was 95.83%, and the accuracy rate was 77.38% (Figure 1).

Table 1 shows the general characteristics of the 60 patients. Forty-two patients (average 65.5 ± 1.69 years of age) with a correct diagnosis by ROSE were compared with 18 patients (average 65.5 ± 2.38 years of age) with an incorrect diagnosis by ROSE. The mean size of the sampled mediastinal LNs in the two groups was 24.55 ± 2.23 vs. 22.86 ± 4.13 mm (*p* = 0.698). The mean number of mediastinal lymph node stations biopsied per patient was 1.7 (range: 1.0–2.3), and the mean FDG uptake in the two groups was 10.77 ± 0.75 vs. 6.45 ± 0.78 (*p* = 0.001). In the two groups, a SUVmax of mediastinal LNs > 5 (*p* < 0.001) and a mediastinal LN size > 1.5 cm (*p* = 0.027) was statistically significant. No statistical significance was observed for age (*p* = 1), gender (*p* = 0.678), smoking history (*p* = 0.78), histology (*p* = 0.335), tumor differentiation (*p* = 0.145), EGFR mutation (*p* = 0.321), Hounsfield units (HU) (*p* = 0.852), ROSE slides (*p* = 0.594), carcinoembryonic antigen (CEA) level (*p* = 0.695), squamous cell carcinoma antigen level (*p* = 0.515), or survival (*p* = 0.309). None of the patients had endobronchial mucosal abnormalities. All EBUS-TBNA procedures were performed under general anesthesia by two experienced pulmonologists. A median of three passes was performed for each node (range: 2–5).

### 3.2. Univariate and Multivariate Analysis of Predictive Factors of the Diagnostic Accuracy of ROSE in EBUS-TBNA in Non-Small-Cell Lung Cancer Patients with Malignant Mediastinal LNs

Table 2 shows the results of the univariate regression and multivariate regression analyses. In the univariate analysis, mediastinal LNs larger than 1.5 cm (HR = 3.667, *p* = 0.031) and a SUVmax greater than 5 (HR = 41, *p* = 0.001) were statistically significant for the diagnostic accuracy of ROSE in EBUS-TBNA. However, according to the multivariate Cox regression analysis, only a SUVmax greater than 5 (HR = 20.258, *p* = 0.016) was statistically significant.

### 3.3. Survival Analysis

Kaplan–Meier analysis revealed no significant differences (*p* = 0.418) in the 3 year overall survival (OS) between patients with a correct diagnosis and those with an incorrect diagnosis by ROSE (Figure 2).

## 4. Discussion

### 4.1. The Role of ROSE in Clinical Practice

Precise assessment of lymph node metastasis is important for selecting the optimal treatment for patients with NSCLC. Pathologic diagnosis of LNs is generally considered credible. EBUS-TBNA is a relatively non-invasive procedure compared with mediastinoscopy and video-assisted thoracoscopic surgery with lymph node dissection. The role of ROSE in EBUS-TBNA is to verify sample adequacy and to establish a preliminary diagnosis by performing a rapid stain in the bronchoscopy suite or operating room, followed by evaluation by a cytopathologist or a trained cytotechnologist [7,15]. The utility of ROSE has been critically examined in the literature [4,16]. Therefore, ROSE is believed to be a tool that increases the adequacy rate, diagnostic yield, and accuracy of EBUS-TBNA.

Theoretically, ROSE is not needed because EBUS is a real-time technique in which the needle and target may be observed during the procedure to verify the procedure’s accuracy [6]. Other arguments include the need for an experienced dedicated cytopathologist as a limitation of ROSE because all hospitals and institutes may not be able to accommodate this requirement [6]. Other criticisms of ROSE include the uncertainty of the diagnostic material in a paraffin block, despite the use of ROSE services, the time spent in performing the evaluation of repeated staining and sample analysis, and the cost [17]. The staining method employed during ROSE, however, does not influence the final diagnosis [18]. Moreover, on the basis of its convenience and efficiency in lung cancer patients, we typically perform EBUS-TBNA for nodal staging followed by surgical resection of lung cancer without nodal metastases according to the result of ROSE. Therefore, the diagnostic accuracy of ROSE plays a particularly important role in clinical practice, especially in nodal staging used for the selection of subsequent treatment, including surgical resection or systemic therapy.

In our study, the correlation rate between correct diagnosis by ROSE and correct diagnosis by final pathology was 71.43% (*p* = 0.003). To the best of our knowledge, no associated studies have investigated predictors of the diagnostic accuracy of ROSE in EBUS-TBNA. We hypothesized that the result of ROSE in selected patients without predictors should be carefully considered, at which point treatment decisions for patient management may be made.

### 4.2. Predictive Factors of the Diagnostic Accuracy of ROSE in EBUS-TBNA

To the best of our knowledge, the predictive factors of preoperative lymph node metastasis are central tumor localization [19], larger tumor size [19,20,21], age ≤ 67 years [20], high CEA level [20,21], micropapillary-predominant adenocarcinoma [21,22,23], and consolidation/tumor ratio ≥ 89% [20]. Moreover, although EGFR mutations are strongly associated with clinical outcomes in patients with lung adenocarcinoma, the effect of EGFR mutation status on prognosis is still unclear [24,25,26,27,28,29]. In our study, age, gender, smoking history, histology, tumor differentiation, EGFR mutation status, CEA level, and OS were not statistically significant factors related to the diagnostic accuracy of ROSE.

In the univariate regression analysis, the SUVmax and size of mediastinal LNs were significant predictors of the diagnostic accuracy of ROSE. A SUVmax > 5 (area = 0.894) and mediastinal LN size > 1.5 cm (area = 0.643) as cutoffs were determined by the area under the receiver operating characteristic curve. After the multivariate analysis, a SUVmax > 5 was the only independent predictor of the diagnostic accuracy of ROSE in EBUS-TBNA. High FDG uptake, as seen by PET, that is suspicious for nodal metastases requires histopathologic confirmation and represents higher metabolic activity [30]. FDG-PET/CT adds greater value to CT in lymph node staging of NSCLCs [31]. Therefore, the density and behavior of malignant cells in mediastinal LNs could be correlated with the FDG uptake value. In clinical practice, we perform EBUS-TBNA and possible surgical resection of lung cancer at the same time when patients with suspected mediastinal nodal metastases are under general anesthesia. The diagnostic accuracy of ROSE is valuable and helps direct immediate decisions regarding subsequent disease management. Therefore, in clinical practice, we could use predictors of ROSE accuracy to consider the results and make rational decisions. We also noted that the size of the mediastinal LNs corresponds with the volume of the available specimen. Thus, patients with larger LNs benefited from a higher diagnostic accuracy of ROSE and received more needle aspirations per target lesion in our study. A SUVmax less than 5 should be considered a risk factor of lower diagnostic accuracy of ROSE.

One limitation of this study was its retrospective single-center design. Additionally, the small sample size may have led to bias. A prospective, randomized, multicenter study is necessary to confirm the predictive accuracy of ROSE during EBUS-TBNA.

## 5. Conclusions

A SUVmax greater than 5 is an independent predictor of higher diagnostic accuracy of ROSE in EBUS-TBNA in NSCLC patients with metastatic mediastinal LNs. The patients with larger LNs benefited from a higher diagnostic accuracy of ROSE and received more needle aspirations per target lesion in our study. In clinical practice, the utility of ROSE in patients with certain risk factors that might affect the diagnostic accuracy should be carefully evaluated and requires further prudent assessment (more shots or repeated biopsies at mediastinal LNs).

## Figures and Tables

**Figure 1 jpm-12-00451-f001:**
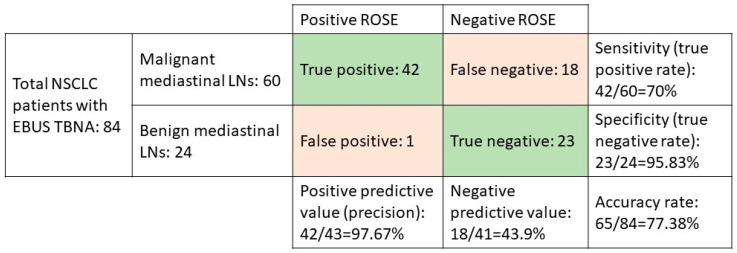
The confusion matrix of NSCLC patients who underwent EBUS-TBNA.

**Figure 2 jpm-12-00451-f002:**
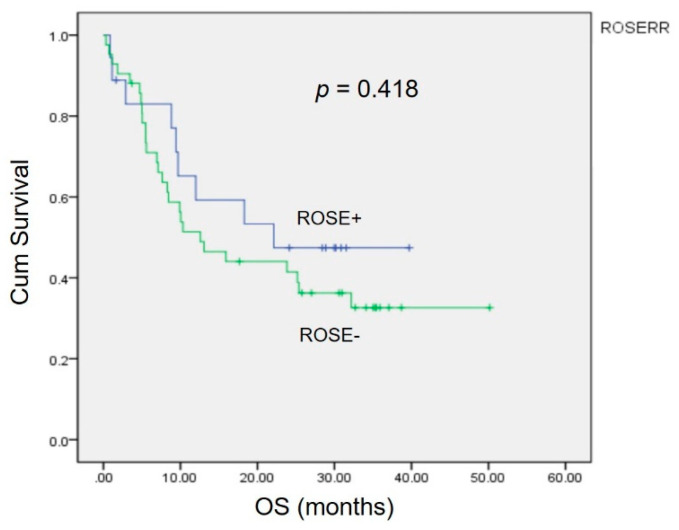
The impact of the accuracy of ROSE in EBUS on overall survival. The Kaplan–Meier curve showed no statistically significant difference (*p* = 0.418).

**Table 1 jpm-12-00451-t001:** Characteristics of non-small cell lung cancer patients with malignant cells involved in mediastinal lymph node by endobronchial ultrasound with trans-bronchial needle biopsy.

	Correct Diagnosis of ROSE n = 42 (%)	Incorrect Diagnosis of ROSE n = 18 (%)	*p*-Value ^a^
Age	65.5 ± 1.69	65.5 ± 2.38	1
Gender			0.678
Male	28 (66.67)	13 (72.22)	
Female	14 (33.33)	5 (27.78)	
Smoking			0.78
Yes	24 (57.14)	11 (61.11)	
No	18 (42.86)	7 (38.89)	
Histology			0.335
Adenocarcinoma	28 (66.67)	5 (27.78)	
SCC	7 (16.67)	3 (16.67)	
PDA	5 (11.9)	4 (22.22)	
Clinical stage			0.875
IA	1 (2.38)	0	
IB	0	1 (5.56)	
IIB	3 (7.14)	0	
IIIA	9 (21.43)	5 (27.78)	
IIIB	8 (19.05)	4 (22.22)	
IVA	6 (14.29)	5 (27.78)	
IVB	15 (35.71)	3 (16.67)	
Differentiation			0.145
Moderate	15 (35.71)	3 (16.67)	
Poor	27 (64.29)	15 (83.33)	
EGFR			0.321
Mutation	11 (39.29)	3 (23.08)	
Wild-type	17 (60.71)	10 (76.92)	
SUVmax of mediastinal LNs	10.77 ± 0.75	6.45 ± 0.78	0.001 ^a^
SUVmax of mediastinal LNs > 5			<0.001 ^a^
Yes	37 (88.10)	12 (66.67)	
No	2 (4.76)	4 (22.22)	
Mediastinal LN size (cm)	24.55 ± 2.23	22.86 ± 4.13	0.698
Mediastinal LN size > 1.5 cm			0.027 ^a^
Yes	33 (78.57)	9 (50)	
No	9 (21.43)	9 (50)	
Hounsfield units (HU)	54.14 ± 2.81	53.18 ± 4.18	0.852
ROSE slides	6.62 ± 0.357	6.28 ± 0.497	0.594
CEA (ng/mL)	47.4 ± 26.39	29.46 ± 12.17	0.695
Anti-SCC (ng/mL)	2.94 ± 1.43	1.3 ± 0.37	0.515
Final pathology of EBUS			0.003 ^a^
Correct	30 (71.43)	8 (50)	
Incorrect	4 (9.52)	8 (50)	
Operation time (min)	50.4 ± 3.23	64 ± 6.42	0.04 ^a^
Survival			0.309
Yes	15 (35.71)	9 (50)	
No	27 (64.29)	9 (50)	

^a^ Significance was assessed using χ^2^ tests. Key: EBUS, endobronchial ultrasound; ROSE, rapid on-site evaluation; SCC, squamous cell carcinoma; PDA, poorly differentiated carcinoma; EGFR, epidermal growth factor receptor; SUV, standard uptake value; N2 LN, mediastinal lymph node at pretracheal retrocaval space or subcarinal area; CEA, carcinoembryonic antigen.

**Table 2 jpm-12-00451-t002:** Univariate and multivariate logistic regression of predictors for diagnostic accuracy of ROSE.

	Univariant		*p*-Value ^a^	Multi-Variant		*p*-Value
	HR	CI (95%)		HR	CI (95%)	
N2 LN size > 1.5 cm	3.667	1.125–11.955	0.031 ^a^	1.867	0.278–12.537	0.521
Pathology accuracy	7.5	1.792–31.383	0.006 ^a^	1.548	0.171–13.986	0.697
SUVmax > 5	41	4.596–365.734	0.001 ^a^	20.258	1.761–233.057	0.016 ^a^

^a^ Significance was assessed using Student’s *t*-tests. Key: HR, hazard ratio; CI, confidence interval; SUVmax, maximum standard uptake value of FDG; N2 LN, mediastinal lymph node 4 or 7.

## Data Availability

The datasets generated and analyzed during the current study are available from the corresponding author on reasonable request.

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
