# Peer review of "High SUVmax Is an Independent Predictor of Higher Diagnostic Accuracy of ROSE in EBUS-TBNA for Patients with NSCLC"

_jpm, 2022, doi:10.3390/jpm12030451_

Round 1

Reviewer 1 Report

It is my honor to review the manuscript entitled 'High SUVmax is an independent predictor of higher diagnostic accuracy of ROSE in EBUS-TBNA in patients with NSCLC'.  However, It is strongly required to clarify the study goal of this manuscript. Do you insist that the L/N with SUV >5 do not need additional biopsy? Otherwise only L/N with SUV >5 must be biopsied?

  1. A complete revision of the analytical method is required for the based on the revised purpose. 
  2. When describing a test method, there is a standardized metric for this. If we are to evaluate the accuracy of the EBUS + SUV value, we need to describe an additional and exhaustive metric.
  3. Please clarify the gold standard of the positive result.

Author Response

Thanks for your kind comments and suggestions.

Question 1. Do you insist that the L/N with SUV >5 do not need an additional biopsy? Otherwise only L/N with SUV >5 must be biopsied?

Answer 1. The aim of our study is to find the predictors of failed rapid on-site evaluation (ROSE) in endobronchial ultrasound with transbronchial needle biopsy (EBUS TBNA). Among our database, we found that the group of L/N with SUVmax > 5 got higher accuracy in ROSE with statistical significance. Therefore, we suggested more shots or repeated biopsies at mediastinal LNs in the group of L/N with SUVmax < 5 because of less accuracy in ROSE. So, EBUS TBNA still can be performed in all patients with suspicion of positive N2 lesions.

Question 2. A complete revision of the analytical method is required based on the revised purpose.

Answer 2. In our study, we enrolled all NSCLC patients who received EBUS TBNA and further investigated the predictors of ROSE of EBUS TBNA in the malignant N2 patients. And, we did not investigate patients with benign N2 because of significantly lower SUVmax. Therefore, we did not suggest changing the study purpose. Thanks for your suggestion.

Question 3. When describing a test method, there is a standardized metric for this. If we are to evaluate the accuracy of the EBUS + SUV value, we need to describe an additional and exhaustive metric.

Answer 3. Thanks for your suggestions. We revised figure 1 as a confusion matrix to explain the precision (positive predictive rate), accuracy, and true positive rate. We added the confusion matrix showed the positive predictive rate was 97.67%, the sensitivity was 70%, the specificity was 95.83%, and the accuracy rate was 77.38%.

Question 4. Please clarify the gold standard of the positive result.

Answer 4. Thanks for your kind suggestions. In the literature review, there is no gold standard value of SUVmax in NSCLC patients for EBUS TBNA. Our study is the pilot study to investigate the importance of SUVmax to predict the diagnostic accuracy of ROSE in EBUS TBNA.

Reviewer 2 Report

 The study entitled “High SUVmax is an independent predictor of higher diagnostic accuracy of ROSE in EBUS-TBNA in patients with NSCLC” by Chen at al., provides comprehensive assessment of rapid-onsite evaluation’s diagnostic accuracy in predicting the correct disease staging by endobronchial ultrasound-guided-transbronchial needle aspiration in NSCLC patients. The study provides an interesting insight into the factors i.e., LN size and SUVmax deciding the accuracy of ROSE. As mentioned, the small sample size is one of the limitations of the study, but the manuscript is well written.

I have the following concerns:

  1. Please correct the referencing format in introduction. Follow one format throughout the manuscript.
  2. The Figure 1 can be further improved by mentioning more details.
  3. In the material and methods, the mediastinal lymph node sampling mentions the number of passes was approximately twenty…Please check for corrections.
  4. In Table 1 the p value for clinical stage column is missing. Please add.
  5. Also in Table 1 please correct the spacing error for survival column by aligning yes and no.
  6. Please check for alignment and formatting issues of text throughout the manuscript. Follow one text size.

Author Response

Thanks for your kind comments and suggestions.

Question 1. Please correct the referencing format in the introduction. Follow one format throughout the manuscript.

Answer 1. Thanks for your suggestions. We have revised these errors.

Question 2. Figure 1 can be further improved by mentioning more details.

Answer 2. Thanks for your kind suggestions. We have revised it as a confusion matrix for better explanation.

Question 3. In the material and methods, the mediastinal lymph node sampling mentions the number of passes was approximately twenty…Please check for corrections.

Answer 3. Thanks for your suggestions. We revised it as 20 to 30 times.

Question 4. In Table 1 the p-value for the clinical-stage column is missing. Please add.

Answer 4. Thanks for your suggestion. We have revised it.

Question 5. Also in Table 1 please correct the spacing error for the survival column by aligning yes and no.

Answer 5. Thanks for your suggestions. We have revised these errors.

Question 6. Please check for alignment and formatting issues of text throughout the manuscript. Follow one text size.

Answer 6. Thanks for your suggestions. We have revised it and corrected the format throughout the manuscript. Thank you very much.
